# Yield prediction for crops by gradient-based algorithms

**Pavithra Mahesh, Rajkumar Soundrapandiyan** [ID] *

School of Computer Science and Engineering, Vellore Institute of Technology, Vellore, Tamilnadu, India

* rajkumars@vit.ac.in

**Data Availability Statement:** All data files are available from the https://www.kaggle.com/code/kushagranull/crop-yield-prediction.

**Funding:** The authors received no specific funding for this work.

## Abstract

A timely and consistent assessment of crop yield will assist the farmers in improving their income, minimizing losses, and deriving strategic plans in agricultural commodities to adopt import-export policies. Crop yield predictions are one of the various challenges faced in the agriculture sector and play a significant role in planning and decision-making. Machine learning algorithms provided enough belief and proved their ability to predict crop yield. The selection of the most suitable crop is influenced by various environmental factors such as temperature, soil fertility, water availability, quality, and seasonal variations, as well as economic considerations such as stock availability, preservation capabilities, market demand, purchasing power, and crop prices. The paper outlines a framework used to evaluate the performance of various machine-learning algorithms for forecasting crop yields. The models were based on a range of prime parameters including pesticides, rainfall and average temperature. The Results of three machine learning algorithms, Categorical Boosting (CatBoost), Light Gradient-Boosting Machine (LightGBM), and eXtreme Gradient Boosting (XGBoost) are compared and found more accurate than other algorithms in predicting crop yields. The RMSE and $R^2$ values were calculated to compare the predicted and observed rice yields, resulting in the following values: CatBoost with 800 (0.24), LightGBM with 737 (0.33), and XGBoost with 744 (0.31). Among these three machine learning algorithms, CatBoost demonstrated the highest precision in predicting yields, achieving an accuracy rate of 99.123%.

## Introduction

"Farming though hard is foremost trade, Men ply at will but ploughmen lead" by saint poet Thiruvalluvar in his Tamil literature "Kural 1031". Agriculture refers to hunger, field, and cultivation. It is the largest livelihood provider in rural India and contributes significantly to the Gross Domestic (GDP) and economy of any country.

Data provided by National Statistical Office (NSO), shows that the share of agriculture and allied sectors in Gross Value Added (GVA) of India has been increasing (Fig 1).

This indicates the significant contribution of the agriculture sector to India's economy. Due to the importance of agriculture, various supportive measures have been implemented to

**Competing interests:** The authors have declared that no competing interests exist.

**Fig 1. Trend of GVA in India during the years 2018–21.**

address the issues faced by the agriculture sector. These measures include government policies and programs, research and development initiatives, technological advancements, etc. These measures intend to improve agricultural productivity, increase the income of farmers, and promote sustainable agricultural practices. These initiatives are also to ensure food security, reduce poverty and promote sustainable development.

## Problem statement

The problem for crop prediction involves addressing the challenges associated with accurately forecasting crop yields in agriculture. Farmers face uncertainties in optimizing their crop selection and planning due to various factors such as environmental conditions, including temperature, soil fertility, water availability, and seasonal variations. Economic considerations, like market demand, purchasing power, and crop prices, further complicate decision-making.

The lack of precise crop yield predictions can lead to financial losses, inefficient resource utilization, and difficulties in strategic planning. To tackle this issue, there is a need for advanced predictive models, specifically leveraging machine learning algorithms. These models should consider a range of parameters such as pesticides, rainfall, and average temperature to provide reliable and timely predictions.

The problem statement aims to explore and develop a solution that enhances the accuracy of crop yield predictions. The focus is on leveraging machine learning techniques to analyze historical and real-time data, ultimately assisting farmers in making informed decisions about crop selection, resource allocation, and overall farm management. The goal is to empower farmers with a tool that minimizes uncertainties, improves crop yield forecasts, and contributes to the overall efficiency and sustainability of agricultural practices.

The primary obstacle that must be overcome to achieve the required demand for food products and improve the welfare of farmers is to enhance agricultural productivity by embracing cutting-edge technologies such as the Internet of Things (IoT), Artificial Intelligence (AI), Data Mining, Neural Networks (NN), etc. Hence a framework to appraise critical farming

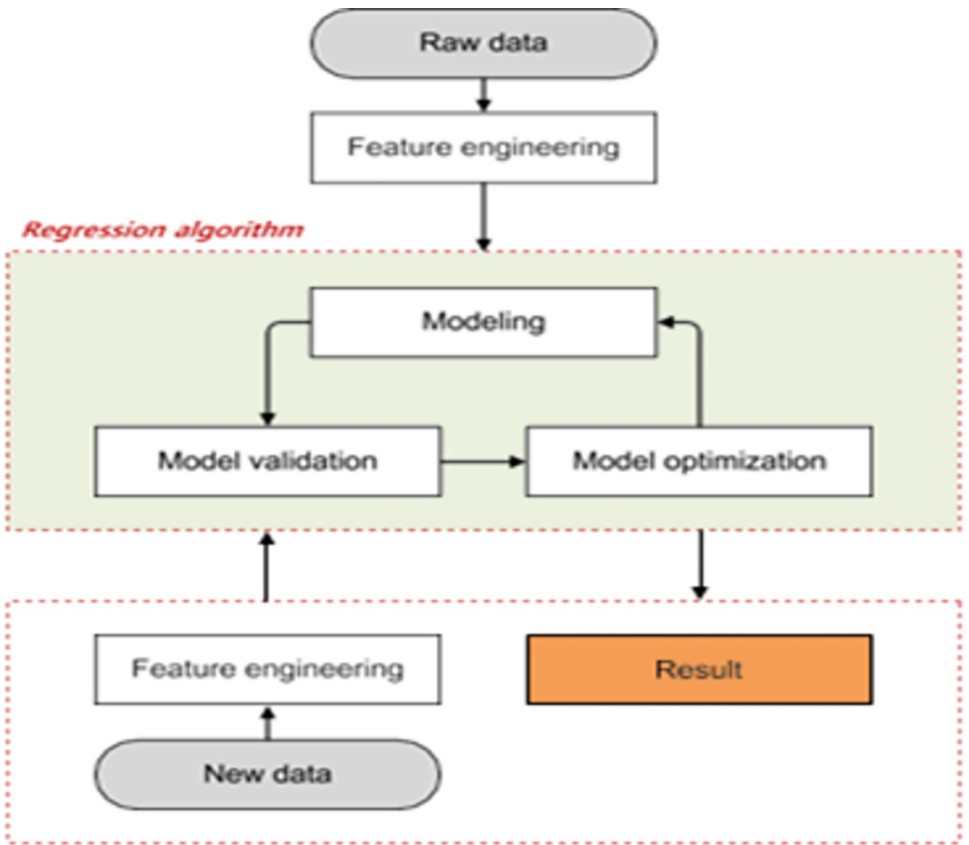

**Fig 2. Proposed framework for yield prediction.**

decisions, huge data need to be gathered and analyzed. This has become much possible with the assistance of modern-day technologies greatly influenced by IoT [1], and data analysis [2–4] that includes data science and data analytics. These latest technologies have enhanced the ability of people in understanding these factors, assist decision-making, and help to choose a suitable crop to achieve good yield. The proposed model is depicted in (Fig 2).

The IoT holds great potential for monitoring and management in the field of agriculture. With the aid of sensors and other devices, farmers can gather important data on a range of factors including soil moisture, temperature, humidity, and crop growth. Analyzing this data enables them to make choices regarding irrigation schedules, fertilization, and crop harvesting, potentially resulting in increased crop yield and efficiency.

IoT is being used in agriculture around the world. Farmers in Australia are using IoT to monitor soil moisture levels and adjust irrigation systems accordingly, while farmers in the US are using IoT to track the movement of livestock and monitor crop health.

However, IoT in agriculture also presents some challenges, such as the need for reliable connectivity in rural areas and the potential cost of the necessary equipment and infrastructure. The advancement in technology has aided to overcome these challenges. Farmers can leverage suitable sensors to collect data related to agriculture. Acquired data can be remotely analyzed by farmers themselves or researchers by utilizing cloud computing. In addition, data mining can be a powerful tool for extracting valuable insights and patterns from large agricultural datasets. By utilizing data mining techniques to analyze the data collected from IoT

devices, farmers and researchers can identify significant trends and patterns that can inform their decision-making processes around crop planting, irrigation, fertilization, and harvesting.

Data analysis [5] is a critical part of the data mining process. It involves pre-processing the data, selecting suitable algorithms to analyze the data, and generating useful knowledge and conclusions. This can help farmers and researchers to identify information for the decision-making process. The implementation of IoT devices and data mining techniques can significantly improve crop management for farmers and researchers. This can lead to enhanced crop yield, improved efficiency, and increased profitability within the field of agriculture.

Manual search and analysis of huge data will be a difficult task. Automated applications adopting AI techniques will be better and more supportive. AI encompasses machine learning. The integration of machine learning into agriculture has the potential to transform the industry by providing farmers with valuable decision-making tools and insights based on data analysis. This can lead to enhanced crop yields, operational efficiency, and profitability, while also decreasing the environmental impact of farming practices.

The motivation of the study is to develop an effective machine learning-based approach to crop yield prediction. Model a Prediction tool based on accurate crop yield predictions that can assist farmers and decision-makers in crop management, resource allocation, and risk mitigation strategies. With the uncertain weather patterns due to global warming, such a tool can be particularly useful in helping farmers adapt to changing conditions and ensure food security for the growing population.

The main contribution of the work presented in this paper is outlined below

- Leveraging publicly available data on weather, agricultural practices, pesticides, and chemicals, a predictive model capable of accurately forecasting crop yields in India has been developed.

- One-Hot Encoding has been used to convert categorical variables to the one-hot numeric array.

- Three different machine learning algorithms (CatBoost, LightGBM, and XGBoost) have been adopted in the model for achieving accurate prediction results for crops.

- The developed robust prediction framework has been modeled to effectively avoid overfitting and underfitting scenarios.

The subsequent sections of the paper delve into specific facets, exploring the application of gradient methods in crop recommendation. The focus on investigating the performance of gradient based machine learning models, coupled with discussions on model-building times, sets the stage for the results section. Additionally, the paper outlines future directions, envisioning an integrated framework and continuous environmental monitoring. Through these comprehensive discussions, the paper strives to offer a holistic perspective on the intersection of framing, IoT, and data analysis in the context of precision agriculture. The findings presented in this paper shall provide a platform for researchers to further enhance and develop a crop yield prediction application tool.

The paper is organized as follows. The work detailed is focused on using machine learning techniques to predict crop yield. Section 1 provides an overview of related work in this area, while Section 2 describes the study area, crops, data source, mathematical model, and parameterization. Section 3 discusses the experimental results and Section 4 concludes the paper with some perspectives. It would be interesting to learn more about the specific machine-learning models used in this study and how they were trained and evaluated.

## Section 1: Related work

Suitable crops [6] can be decided based on the soil properties and atmospheric conditions for the specific area to have better harvesting. A framework for the selection of the best suitable crop according to farmland is presented [7, 8]. The study [9] evaluated the efficacy of Random Forest (RF) regression using Multi Linear Regression (MLR) as a benchmark to model complex yield responses of wheat, grain maize, potato, and silage maize at global and regional scales. Machine learning [10–12] can be applied for solving to three main types of problems. In [13] Firefly-XGBoost model emerges as an innovative solution designed specifically for forecasting and reconciling blasting outcomes, with a particular focus on mean fragmentation size (MFS) and peak particle velocity (PPV). The enhancement of the final yield prediction's timeliness and robustness can be achieved by integrating the crop mechanism model with a statistical regression model (SRM) in the crop yield prediction system. This research specifically incorporates accumulated biomass (AB), simulated by the Agricultural Production Systems SIMulator (APSIM) model, along with various climate indices such as climate suitability indices and extreme climate indices, into SRM for predicting wheat yield in the North China Plain (NCP). The outcomes reveal that utilizing the random forest (RF) algorithm in the prediction model yields favorable results. [14]. In research work presented in [15] introduces a soil liquefaction prediction framework utilizing a relatively recent and resilient class of tree-based ensemble algorithms, namely Adaptive Boosting, Gradient Boosting Machine, and eXtreme Gradient Boosting (XGBoost), applied to the Standard Penetration Test (SPT) dataset. In [16] experimentation has been done adopting ML methods, including ANN, RF, XGBoost, SVM, and MLR models to forecast air quality in Macau during air pollution episodes. The work [17] proposed a novel AI based approach to diabetes classification. XGBoost approach has been used to process large amounts of data at a relatively quick rate. In [18] credit card fraud detection was modeled with the XGBoost algorithm. You et al. [19] proposed a deep learning framework that combined remote sensing data, soil data, and climate data to predict crop yields for most developing countries throughout the year. Paudel et al. [20] proposed a framework that utilized machine learning algorithms and crop modeling principles to predict large-scale crop yields. Sun et al. [21] utilized a multilevel deep learning model that combined RNN and CNN for extracting spatial and temporal features from remote sensing and soil property data to predict crop yield in the US Corn Belt. Similarly, Shahhosseini et al. [22] conducted an investigative study that demonstrated the impact of integrating crop modeling and machine learning on improving yield prediction. Shook et. al. [23] used performance records from Uniform Soybean Tests in North America to build a Long Short-Term Memory (LSTM). The proposed framework outperformed other machine learning models such as Support Vector Regression with Radial Basis Function kernel, least absolute shrinkage and selection operator regression, and the data-driven USDA model for yield prediction.

The research publications provide evidence that machine learning through integration of remote sensing data, soil and weather data, and crop modeling principles, has the potential to do accurate crop prediction. The framework assists farmers and policymakers in making decisions regarding crop management practices and enhancing food security amidst environmental adversities.

## Section 2: Materials and methods

The terms/materials used for presenting the proposed framework are briefed for improving readability and better clarity. The overview of the working model of the proposed system is shown in (Fig 3). The user provides the location and month as input. The system retrieves the latitude and longitude coordinates corresponding to the given location and integrates them

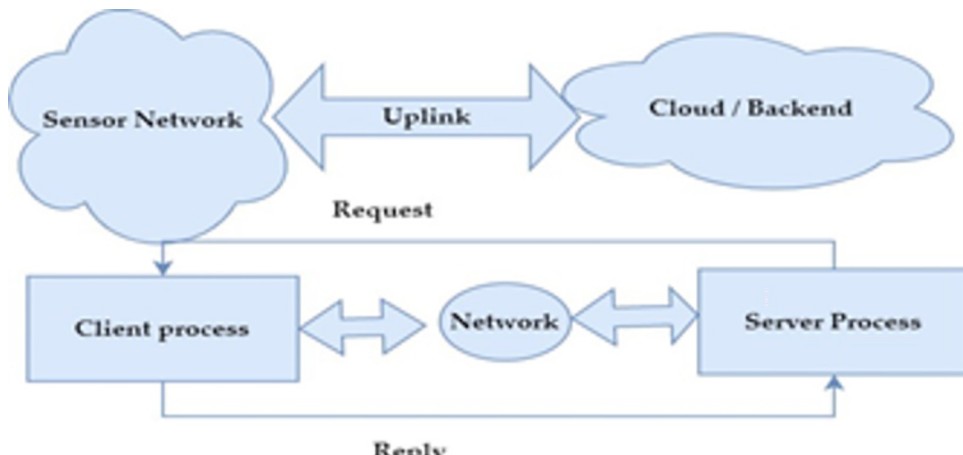

**Fig 3. Overview of the proposed model.**

with real-world data sourced from storage media, which is then stored in a cloud-based database management system. After receiving the request, the trained model recommends one of 10 crop options that are predicted suitable for the available conditions. The block diagram of the proposed model is shown in (Fig 4)

## Dataset and pre-processing

Crop yield prediction is an important agricultural problem. The Agricultural yield primarily depends on weather conditions (rain, temperature, etc), pesticides, and accurate information about the history of crop yield is an important thing for making decisions related to agricultural risk management and future predictions. In this paper, the prediction of the top 10 most consumed yields all over the world is established by applying machine learning techniques. It consists of the 10 most consumed crops namely Cassava, Maize, Plantains, Potatoes, Rice, paddy, Sorghum, Soybeans, Sweet potatoes, Wheat, and Yams. Crops yield of the ten most consumed crops around the world was downloaded from the Kaggle website [24]. The collected data include country, item, year starting from 1961 to 2016, and yield value.

There are two categorical columns in the data frame, categorical data are variables that contain label values rather than numeric values. The number of possible values is often limited to

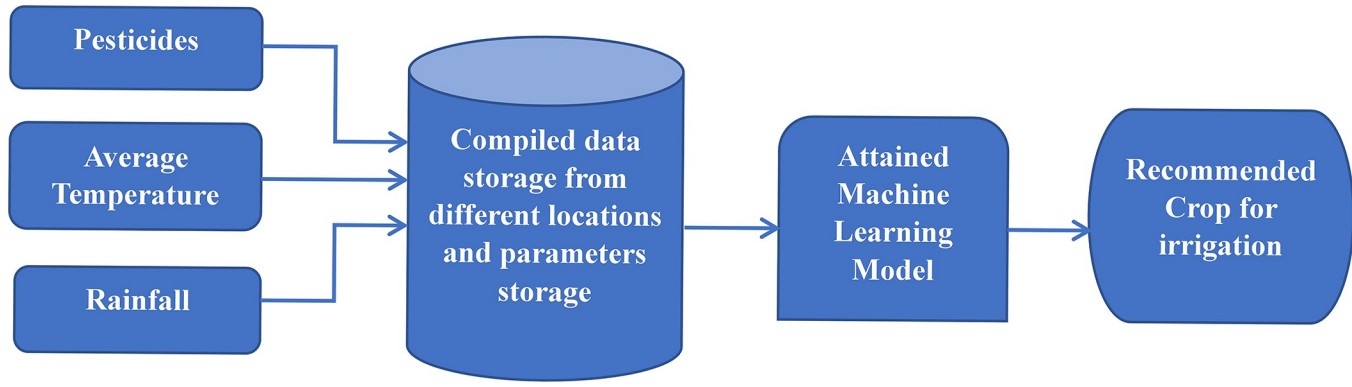

**Fig 4. Block diagram using machine learning module.**

a fixed set, like in this case, items and countries' values. Many machine learning algorithms cannot operate on label data directly. They require all input variables and output variables to be numeric. The categorical data must be converted to a numerical form. One hot encoding is a process by which categorical variables are converted into a form that could be provided as input to ML algorithms to do a better job in prediction. For that purpose, One-Hot Encoding has been used to convert these two columns to the one-hot numeric array.

## Classification for crop yield prediction

A classifier is a ML algorithm that can automatically categorize datasets into predefined classes. It achieves this by training on the features of the data that best represent them and then categorizing unknown data based on its training experience. In the proposed framework, XGBoost [25] and CatBoost [26] algorithms have been implemented as classifiers to perform the prediction task.

**XGBoost.** XGBoost is an effective supervised learning algorithm that leverages gradient boosting to make accurate predictions for a target variable by amalgamating an ensemble of estimates from a group of simpler and weaker models. The algorithm functions by adding weak learners, such as decision trees, to the ensemble iteratively, with each successive model trying to correct the errors of the previous models.

The primary advantages of XGBoost are its capability to handle a diverse array of data types and distributions, including numerical and categorical data. Additionally, the algorithm provides support for an extensive range of hyperparameters that can be fine-tuned to enhance performance. Apart from its flexibility, XGBoost is also incredibly scalable, parallel, distributed, out-of-core, and cache-aware computing features. This enables it to handle large datasets with ease and makes it more than ten times faster than other popular machine learning and deep learning models.

In summary, XGBoost is a well-optimized and scalable algorithm that is particularly suitable for regression, classification (binary and multiclass), and ranking problems. Its ability to handle various data types and distributions, along with its scalability and speed, makes it a suitable option for both machine-learning competitions and real-world applications. The XGBoost algorithm is detailed below:

For K trees, the model is given by Eq (1)

$$\sum_{K=1}^{K} f_k \tag{1}$$

Having all K decision trees, prediction is done by Eq (2)

$$\sum_{K=1}^{K} f_{k\hat{y_i}=\sum_{k=1}^{k} f_k(X_i)} \tag{2}$$

Feature vector $x_i$ belongs to the i-th data point and prediction at the t-th step can be defined as Eq (3)

$$\hat{y}_i^{(t)} = \sum_{k=1}^{t} f_k(X_i) \tag{3}$$

Training the model is done by optimizing (minimizing) the loss function given in Eq (4).

$$L = \frac{1}{N} \sum_{i=1}^{N} \sum_{j=1}^{M} y_{i,j} \log(p_{i,j}) \tag{4}$$

The training process is controlled to avoid overfitting by including a control parameter ($\Omega$)

given in Eq (5) along with the Loss function.

$$\Omega = \gamma T + \frac{1}{2}\lambda \sum_{j=1}^{T} W_j^2 \qquad (5)$$

T denotes the number of leaves with a score on the j-th leaf $W_j$

The final objective function including the control parameter is given in Eqs (6) and (7).

$$Obj = L + \Omega \qquad (6)$$

$$obj = -\frac{1}{N}\sum_{i=1}^{N}\sum_{j=1}^{M} y_{i,j}\log\left(p_{i,j}\right) + \gamma T + \frac{1}{2}\lambda \sum_{j=1}^{T} w_j^2 \qquad (7)$$

The objective function in Eq (7) is responsible for prediction accuracy and the control parameter reduces the complexity of the model.

XGBoost uses gradient descent to optimize its objective function. The algorithm employs an iterative technique that calculates the error in each iteration and moves along the direction of the gradient to minimize the objective function.

Training Objective: The objective function given in Eq (6) is redefined in Eq (8) to suit finding solutions by adopting iterative algorithms.

$$Obj^{(t)} = \sum_{i=1}^{N} L(y_i, \hat{y}_i^t) + \sum_{i=1}^{t} \Omega(f_i) \qquad (8)$$

$$Obj^{(t)} = \sum_{i=1}^{N} L(y_i, \hat{y}_i^t) + \sum_{i=1}^{t} \Omega(f_i)$$

$$= \sum_{i=1}^{N} L(y_i, \hat{y}_i^{t-1} + f_t(x_i)) + \sum_{i=1}^{t} \Omega(f_i) \qquad (9)$$

To optimize Eq (9) by adopting gradient descent, the gradient must be calculated. The performance can be further improved by employing both the first-order gradient given in Eq (10) and the second-order gradient given in Eq (11)

$$\partial_{\hat{y}_i^{(t)}} obj^{(t)} \qquad (10)$$

$$\partial^2_{\hat{y}_i^{(t)}} obj^{(t)} \qquad (11)$$

A simplified version of Eq (9) is obtained by removing the constant terms as calculated in Eq (12)

$$Obj^{(t)} = \sum_{i=1}^{n} \left[ g_i f_i(x_i) + \frac{1}{2} h_i f_i^2(x_i) \right] + \Omega(f_i)$$

$$Obj^{(t)} = \sum_{i=1}^{n} \left[ g_i f_i(x_i) + \frac{1}{2} h_i f_i^2(x_i) \right] + \gamma T + \frac{1}{2}\lambda \sum_{j=1}^{T} w_j^2$$

$$Obj^{(t)} = \sum_{j=1}^{T} \left[ \left(\sum_{i\in I_j} g_i\right) w_j + \frac{1}{2}\left(\sum_{i\in I_j} h_i + \lambda\right) w_j^2 \right] + \gamma T \qquad (12)$$

Where $I_j$ represents the instance of leaf t and the equations to calculate $I_j$, $g_i$, and $h_i$ are given

by Eq (13)

$$I_j = \{i | q(x) = j\}$$

$$g_i = \frac{\partial 1(y_i^{(\hat{t-1})}, y_i)}{\partial (y_i^{(\hat{t-1})})}$$

$$h_i = \frac{\partial^2 1(y_i^{(\hat{t-1})}, y_i)}{\partial (y_i^{(\hat{t-1})})^2} \tag{13}$$

The optimal weight of leaf $j$ and $W_j^*$ can be estimated by Eq (14).

$$W_j^* = \frac{\sum_{i \in I_j} g_i}{\sum_{i \in I_j} h_i + \lambda} \tag{14}$$

A function to be used as a scoring function to measure the quality of a tree structure q, for a given tree structure can be calculated by Eq (15)

For a given tree structure, its quality Obj ($x_i$) shall be calculated by Eq (15).

$$Obj^{(t)}(q) = \frac{1}{2} \sum_{j=1} T \frac{(\sum_{i \in I_j} g_i)^2}{(\sum_{i \in I_j} h_i + \lambda)} + \lambda T \tag{15}$$

Typically, to measure the quality of split nodes by applying scoring in the instance set of left **IL** and right **IR** nodes. The loss reduction after splitting is calculated using Eq (16).

$$Obj_{Split} = \frac{1}{2} \left[ \frac{(\sum_{i \in I_L} g_i)^2}{(\sum_{i \in I_L} h_i + \lambda)} \right] + \frac{(\sum_{i \in I_R} g_i)^2}{(\sum_{i \in I_R} h_i + \lambda)} + \frac{(\sum_{i \in I} g_i)^2}{(\sum_{i \in I} h_i + \lambda)} - \gamma \tag{16}$$

where $I = I_R \cup I_L$

**CatBoost.** Gradient Boosted Decision Trees (GBDT) is a popular technique in supervised machine learning. CatBoost, "CatBoost: unbiased boosting with categorical features" by Prokhorenkova et al. (2018), is a variant of GBDT that introduces two innovative techniques: Ordered Target Statistics and Ordered Boosting. GBDT is a suitable method for learning problems that involve heterogeneous features, noisy data, and complex dependencies, such as web search, recommendation systems, and weather forecasting.

The two innovations introduced by CatBoost aim to address some of the challenges faced during handling different types of data. Ordered Target Statistics involve encoding categorical features based on the mean value of the target variable for each category, which can help handle categorical data more efficiently and accurately. Ordered Boosting, on the other hand, introduces a new way of updating weights during the training process, which helps to avoid overfitting and improves the overall performance of the model. In general, CatBoost has performed well on a variety of tasks involving heterogeneous data and has become a popular choice for machine learning practitioners.

CatBoost and XGBoost are both techniques designed to address the issue of prediction shift resulting from a type of target leakage present in all gradient-boosting algorithms currently in use. However, CatBoost represents an upgraded GBDT toolkit that overcomes problems associated with gradient bias and prediction shift. As such, it offers several advantages over other similar algorithms. A cutting-edge algorithm has been integrated to automatically handle categorical features as numerical ones.

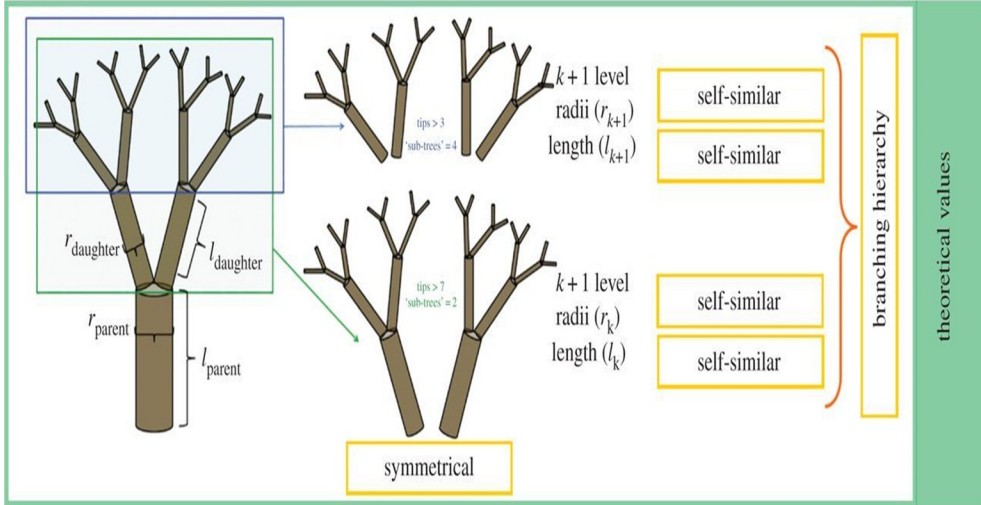

**Fig 5. Symmetric trees at each level.**

By leveraging the interconnections between features through a blend of categorical features, the algorithm substantially enhances the feature dimensions.

To mitigate overfitting and enhance accuracy and generalizability, the algorithm employs a symmetric tree model.

Let $\{(X_k, Y_k)\}$ $nk$-1 be the given input dataset, then $X_i = (x_i,1,\ldots\ldots,x_i,m)$ is a set of m feature vectors and $Y_i \in R$ is a label vector. For simplicity, $x_{i,k}$ is substituted with the value given by Eq (17)

$$\frac{\sum_{j=1}^{n}[X_{j,k} = X_{i,k}].Y_k}{\sum_{j=1}^{n}[X_{j,k} = X_{i,k}]} \tag{17}$$

CatBoost has unique aspects that set it apart from other GBM learning algorithms, particularly in its handling of overfitting problems during training on the entire dataset. Normally, when comparing two variables $X_{i,j}$ and $X_{i,k}$, where [·] equals 1 if $X_{j,k} = X_{i,k}$, and 0 if $X_{j,k} \neq X_{i,k}$.

The approach involves utilizing symmetric trees at each level, as illustrated in (Fig 5), to create a hierarchical tree structure. Identical features are applied to split learning instances at each level. This method incorporates a vectorized representation of the tree, allowing for rapid evaluation.

Incorporating categorical features in unbiased boosting methods can introduce gradient bias issues, which can affect the generalization performance of the resulting model. To address this problem, ordered boosting can be used as a novel approach for gradient estimation, as suggested by [27]. This technique can help improve the generalization ability of the model. In addition, to prevent overfitting, CatBoost employs a random permutation technique based on gradient boosting. This involves randomly arranging the input values to create different permutations, and then computing the average values of the samples for each category.

In the case of a permutation, $\sigma = (\sigma_1, \cdots, \sigma_n)$, it is replaced using Eq (18).

$$\frac{\sum_{j=1}^{p-1}[X_{\sigma_{j,k}} = X_{\sigma_{p,k}}]Y_{\sigma_j} + \beta.P}{\sum_{j=1}^{p-1}[X_{\sigma_{j,k}} = X_{\sigma_{p,k}}] + \beta} \tag{18}$$

where P is a previous value and β is its weight.

**Table 1. Hyper parameters for XGBoost algorithm.**

| Parameter | Hyperparameters |
|---|---|
| Colsample_bytree | 0.67 |
| Max Depth | 17.0 |
| Min_Childweigth | 1.0 |
| Alpha | 57.0 |
| Lambda | 0.9 |

**One-hot encoding.** GBM cannot handle categorical features, which necessitates the use of one-hot encoding. In contrast, CatBoost has integrated an effective encoding process that derives target statistics from categorical features.

**Parameter tuning.** The flexibility of the interface allows for tuning of the hyper-parameters [28], such as the learning rate and tree depth, which are crucial in achieving the highest accuracy in the analysis.

**Parameters selection and settings.** This section describes the selection of booster parameters, learning task parameters, and hyperparameters used for implementing the XGBoost, and CatBoost algorithms.

**XGBoost.** This section provides an overview of XGBoost parameter tuning, hyperparameter tuning, and other valuable information about the algorithm.

The overall parameters have been divided into 3 categories:

*General Parameters*. These parameters formulate the overall functionality of XGBoost. Five parameters fall under this category, the user can set three of them and two are set automatically.

*Booster Parameters*. The proposed model includes both tree booster and linear booster, each with its own set of booster parameters. However, this discussion will focus on the parameters relevant to the tree booster used in the gbtree model.

*Learning Task Parameters*. These parameters are utilized to establish the optimization objective and the metric to be computed at each step.

**CatBoost.** To achieve optimal performance in machine learning models, it is crucial to properly select two specific hyperparameters, maximum depth (max_depth) and learning rate (learning_rate), before beginning the learning process.

To determine the best combination of these hyperparameters, this study utilizes a distributed grid search method, utilizing GridSearchCV from the Python sci-kit-learn library. Table 1 displays the parameters employed for the XGBoost algorithm, (Fig 6) depicts the scores resulting from tuning hyperparameters for CatBoost. The brightest color indicates the optimal combination of the two hyperparameters. In the experimentation of the entire dataset, a maximum depth of 4 is utilized, along with a learning rate of 0.3.

## Section 3: Experimental results

Multiple machine learning models are utilized to train the dataset for predicting crop recommendations, and the optimum model is chosen based on its successful prediction using different attributes from the dataset. The results are validated against a comprehensive database that covers a wide range of soil and environmental parameters. The classification/crop prediction accuracy obtained from experiments using XGBoost and CatBoost models are presented in the following sessions. The results are further compared with those of MLP, Decision tree, and JRip [29].

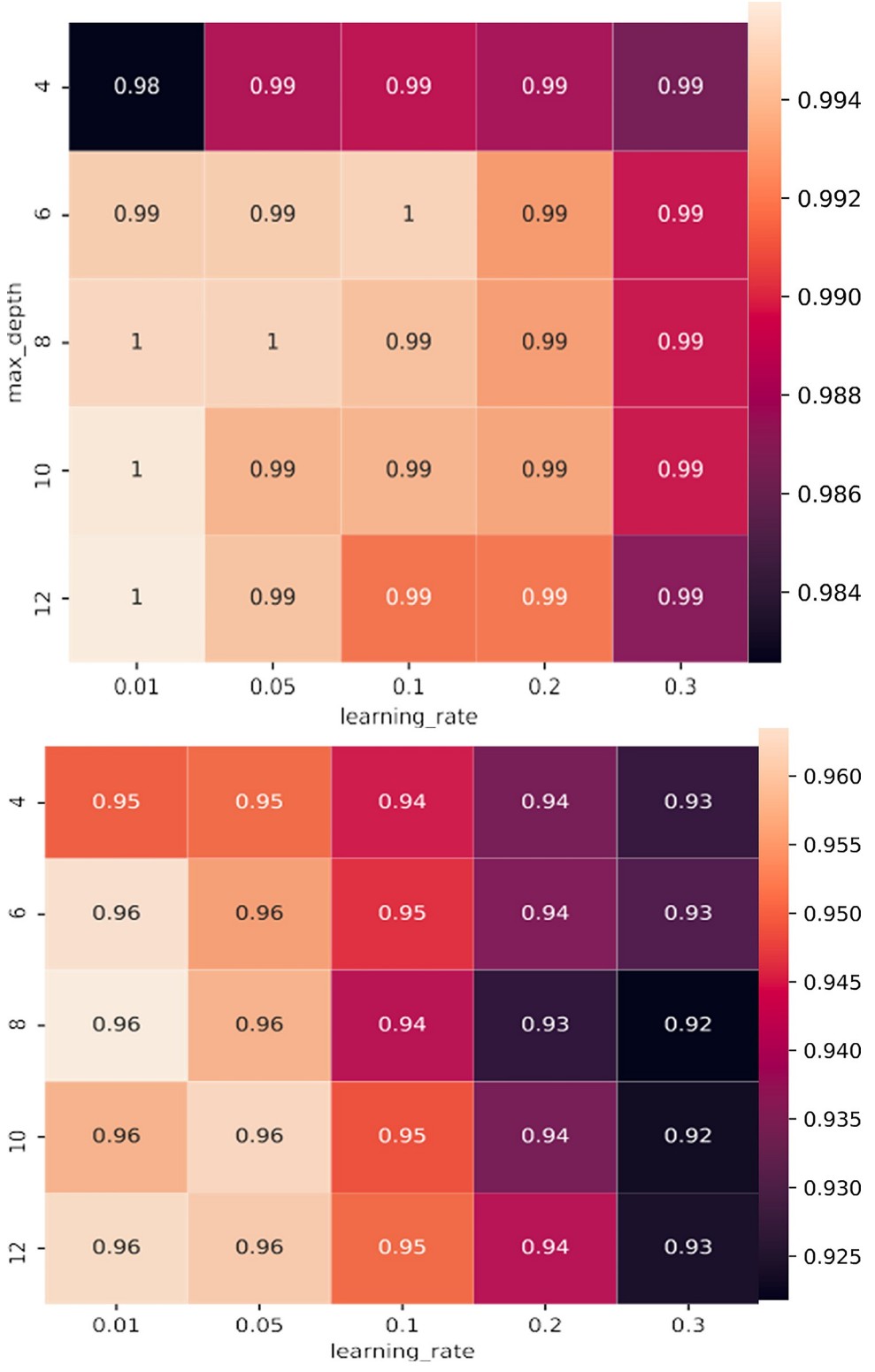

**Fig 6. Hyperparameters for XGBoost and CatBoost algorithms.**

## Evaluation metrics

To evaluate the predictive performance of various models, including CatBoost, XGBoost, LightGBM, CART, AdaBoost, GB, RF, NN, and SVM, comparisons were made using Eqs (19–24) to calculate R2, Mean Absolute Error (MAE), and Root Mean Square Error (RMSE). Additionally, the Diebold Mariano test [30] was conducted to determine if there were significant variations in the predicted accuracy between pairs of competing models, such as CatBoost vs XGBoost, CatBoost vs LightGBM, and XGBoost vs LightGBM

## Coefficient of determination($R^2$)

$$R^2 = \left[ \frac{n \sum ty - (\sum t)(\sum y)}{\sqrt{n(\sum t^2)(\sum t)^2} - \sqrt{n(\sum y^2)(\sum y)^2}} \right]^2 \tag{19}$$

where t is the target value, y is the predicted value and n is the number of samples involved in the evaluation

**Mean Square Error (MSE).**

$$MSE = \frac{1}{n} \sum_{i=1}^{n} (t_i - y_i)^2 \tag{20}$$

**Root Mean Squared Error (RMSE).**

$$RMSE = \sqrt{MSE = \frac{1}{n} \sum_{i=1}^{n} (t_i - y_i)^2} \tag{21}$$

**Mean Absolute Error (MAE).**

$$MAE = \frac{1}{n} \sum_{i=1}^{n} |t_i - y_i| \tag{22}$$

**Coefficient of Variation (CoV).**

$$CoV = \frac{RMSE}{mean} \tag{23}$$

## Mean Absolute Percentage Error (MAPE)

$$MAPE = \frac{1}{n} \sum_{i=1}^{n} \frac{|t_i - y_i|}{t_i} * 100 \tag{24}$$

The dataset is partitioned into 9–1, 8.75–1.25, 8–2, and 7–3 for training and testing purposes. The dataset is evenly split into ten subsets, with 9, 8.75, 8,7 subsets used for constructing a robust learner, while the remaining 1, 1.25, 2, and 3 subsets are employed for model validation. The results depicted in Tables 2 and 3 indicate that all outcomes demonstrate high accuracy, with $R^2$ values nearing unity. Increasing the amount of training data leads to a reduction in RMSE and MAE. In terms of testing the dataset, raising the amount of training data from 80% to 90% enhances $R^2$ from 0.995 to 0.996, while decreasing RMSE and MAE from 280.77

**Table 2. Training dataset portioning results using CatBoost.**

| Data Split | Performance | | | |
|---|---|---|---|---|
| | $R^2$ | MSE | RMSE | MAE |
| 9-1Train | 0.989 | 8093.93 | 89.97 | 8.0517 |
| 9-1Test | 0.996 | 8338.37 | 91.31 | 6.4461 |
| 8.75–1.25 Train | 0.989 | 8153.51 | 90.30 | 6.3952 |
| 8.75–1.25 Test | 0.989 | 8699.34 | 93.27 | 6.3959 |
| 8-2Train | 0.992 | 8586.89 | 92.67 | 3.1798 |
| 8-2Test | 0.994 | 8131.37 | 90.17 | 6.3596 |
| 7-3Train | 0.987 | 8818.13 | 93.90 | 3.1743 |
| 7-3Test | 0.980 | 7997.59 | 89.43 | 6.6753 |

and 119.142 to 231.152 and 109.345, respectively. The findings of the 9–1 test provide the most reliable prediction and are utilized in the subsequent section for evaluation of the proposed framework.

The evaluation of Machine Learning models' accuracy was based on their Mean Absolute Percentage Error (MAPE), with lower MAPE values indicating better performance. The individual MAPE values for XGBoost and LightGBM models are 4.3475 and 2.5097, respectively, while CatBoost achieved the lowest MAPE at 1.5097. The superior performance of CatBoost in terms of the lowest MAPE reaffirms its effectiveness for crop prediction.

**Performance testing.** Prediction models are typically assessed for accuracy using various performance metrics. However, there are instances where conventional evaluation criteria may raise concerns [31]. Despite these situations, many studies employ performance metrics to compare and evaluate the efficacy of algorithms in specific tasks. The prediction accuracy of various models is evaluated using the Diebold-Mariano test [32]. The Diebold-Mariano test (DM) is conducted through the following procedure:

Suppose that $t_t$ and $\hat{y}_{i,t}$ represent the $t^{th}$, $i^{th}$ competing prediction model's actual values and predicted values, respectively.

Subsequently, the $t^{th}$ prediction error of the $i^{th}$ competing prediction model is represented by $e_{i,t}$ (i = 1, 2, 3,..., m), where m is the number of competing models. Eq (25) defines the tth prediction error, $e_{i,t}$.

$$e_{i,t} = t_t - \hat{y}_{i,t} \tag{25}$$

where i = 1, 2, 3,..., m

**Table 3. Training dataset portioning results using XGBoost.**

| Data Split | Performance | | | |
|---|---|---|---|---|
| | $R^2$ | MSE | RMSE | MAE |
| 9-1Train | 0.979 | 10814.10 | 103.99 | 8.0517 |
| 9-1Test | 0.986 | 11399.64 | 106.77 | 6.4461 |
| 8.75–1.25 Train | 0.987 | 10581.99 | 102.87 | 6.9519 |
| 8.75–1.25 Test | 0.987 | 11456.03 | 107.03 | 6.9514 |
| 8-2Train | 0.979 | 11729.00 | 108.30 | 4.3316 |
| 8-2Test | 0.97 | 10690.74 | 103.40 | 4.3473 |
| 7-3Train | 0.987 | 11708.49 | 108.21 | 3.4632 |
| 7-3Test | 0.988 | 7997.59 | 89.43 | 6.6753 |

Table 4. Performance evaluation of CatBoost, XGBoost, and LightGBM by Diebold-Mariano test for comparison.

| Algorithm | MAD | | MSE | | Poly | |
|---|---|---|---|---|---|---|
| | TS | p-value | TS | p-value | TS | p-value |
| CatBoost vs XGBoost | -1.3190 | 0.1874 | -0.9489 | 0.3429 | -0.5897 | 0.5554 |
| CatBoost vs LightGBM | -0.2501 | 0.8025 | -0.8534 | 0.3936 | -1.1626 | 0.2453 |
| XGBoost vs LightGBM | -0.5446 | 0.5871 | -0.7743 | 0.4405 | -0.7823 | 0.4358 |

The accuracy of each prediction model is assessed through the application of loss functions. We gauged the accuracy of predictions using three commonly employed loss functions: Mean Absolute Deviation (MAD), Mean Squared Error (MSE), and a polynomial function that incorporates a power function to assign weights to errors. Employing a single-model approach, the CatBoost model effectively mitigates suboptimal prediction aspects. Table 4 presents the t-statistic (TS) and p-value for various loss functions in CatBoost, XGBoost, and LGBM, utilizing the Diebold-Mariano test for comparison.

Fig 7 shows an illustration of DM test values of CatBoost-XGBoost, XGBoost-LightGBM, and CatBoost-LightGBM. The test determines whether two competing models have different predictive accuracy, it is remarkably observed that the CatBoost paired approaches achieved the best results in terms of p-value, t-statistic compared to the rest of the methods in terms of Diebold Mariano test.

## Performance comparison of XGBoost and CatBoost with several machine learning models

In this study, various machine learning models, such as CART, AdaBoost, GB, RF, LightGBM, NN, and SVM, were compared to XGBoost and CatBoost in terms of their predictive performance. The outcomes, as demonstrated in Table 5, reveal that five boosting ensemble methods (AdaBoost, GB, XGBoost, LightGBM, and CatBoost) and one bagging ensemble method (RF) exhibit high accuracy in prediction. Among these models, CatBoost outperforms the single learning techniques (NN and SVM) in all four evaluation measures. For instance, for Dataset 1, CatBoost increases $R^2$ from 0.962 (NN) to 0.9996, while decreasing MAE from 298.015

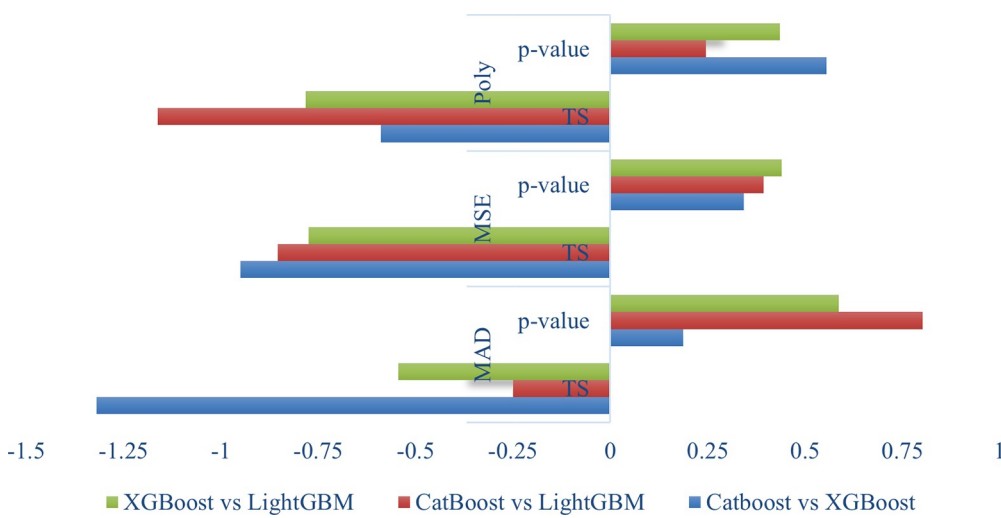

Fig 7. Comparison of TS and p-value of machine learning models using Diebold-Mariano test.

**Table 5. Comparison between the performance of XGBoost and CatBoost with several machine learning models.**

| Algorithm | $R^2$ | MSE | RMSE | MAE |
|---|---|---|---|---|
| CatBoost | 0.990 | 57874.073 | 229.152 | 110.345 |
| XGBoost | 0.984 | 66962.798 | 243.538 | 121.858 |
| CART | 0.978 | 225875.636 | 444.333 | 154.754 |
| AdaBoost | 0.984 | 146579.19 | 372.080 | 249.915 |
| GB | 0.994 | 79287.530 | 267.877 | 109.291 |
| RF | 0.982 | 293742. 469 | 9501.507 | 209.865 |
| LightGBM | 0.990 | 89441.261 | 287.131 | 125.247 |
| NN | 0.962 | 547184.31 | 287.431 | 298.015 |
| SVM | 0.976 | 247092.407 | 488.957 | 286.563 |

(NN) to 109.345, demonstrating a notable performance improvement. Based on these findings, it can be concluded that XGBoost and CatBoost provide the optimal performance with the least amount of error for the examined dataset.

The accuracy of the proposed model using XGBoost and CatBoost was compared with three classifiers, namely function-based and rules-based. The accuracy percentages of the proposed models XGBoost, LightGBM, and CatBoost were found to be 98.40%, 98.72%, and 99.12%, respectively. (Fig 8) shows that the machine learning technique provided an accuracy performance percentage ranging from 98.23%, 88.59% & 96.23% for the multi-layer perceptron (MLP), Decision tree, and JRip classifiers respectively.

To accelerate the classification task, the dataset is preprocessed using normalization. The performance of the classifiers is then validated with the normalized data, and it is verified that the preprocessing of data does not affect the accuracy of the classification models. The results of the models are presented in (Fig 9). However, there is a significant change in the time required to build the model, which is provided in (Fig 10).

## Statistical analysis

When compared to other input parameters in the crop yield dataset, temperature and rainfall feature set have more significant in obtaining higher accuracy in the prediction. We have

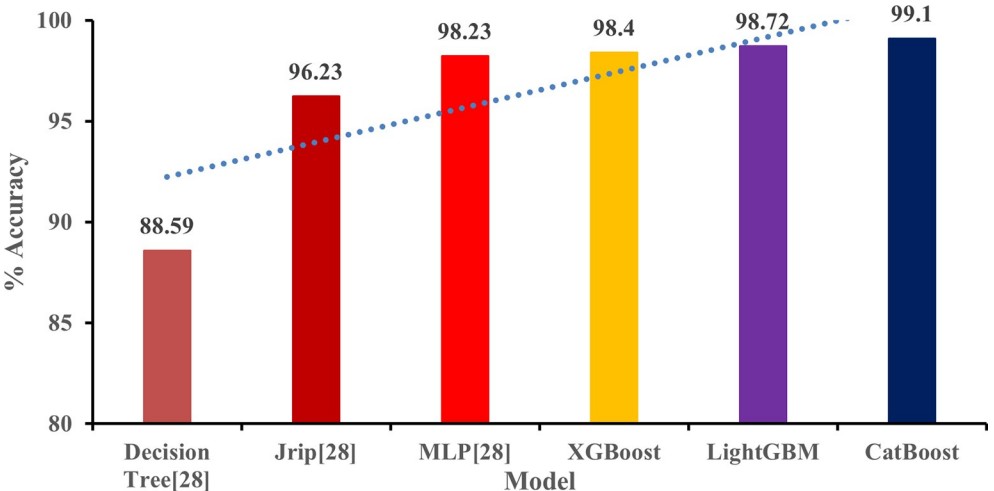

**Fig 8. Classifier versus performance accuracy percentage characteristics.**

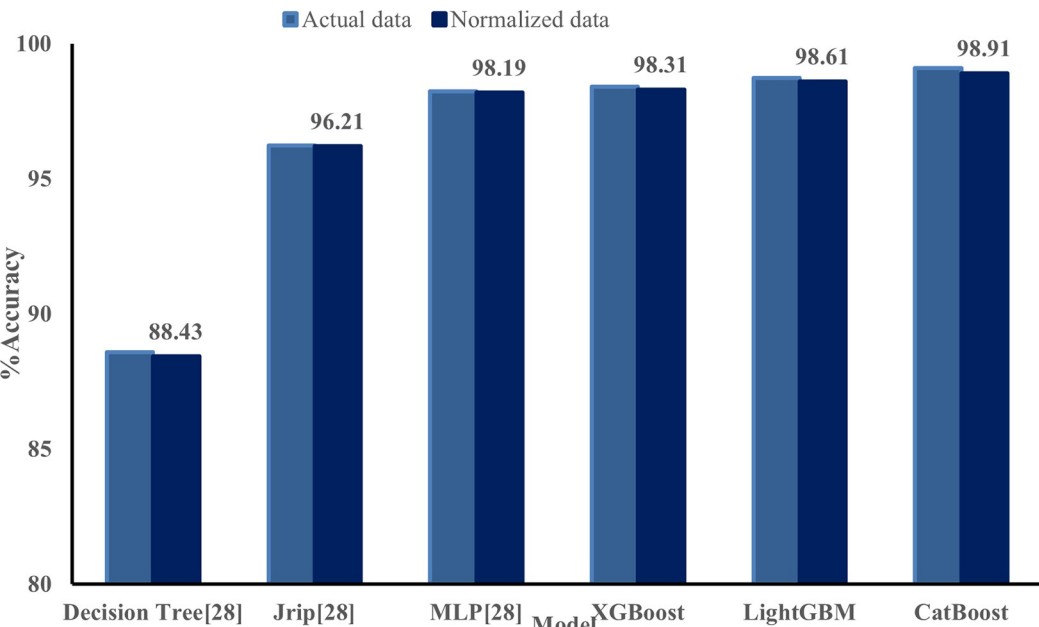

**Fig 9. Comparison of classifier performance using actual and normalized dataset.**

statistically examined and compared pairwise significant differences between the model's outputs from the two feature sets. The three model's numerical outcomes, using two feature sets as inputs. Quantitative performance measure metrics like overall accuracy are used to evaluate the model's ability to predict crop yield.

However, it is necessary to demonstrate statistically that the prediction model's mean results differ from one another. We utilize a one-way ANOVA test because there are more than two models to compare in our investigation.

CatBoost has a smaller variance compared with the other two methods shown in Table 6. As a result, it is evident from the variance values that the CatBoost method outperforms XGBoost and LightGBM approaches in terms of accuracy and dependability. The model's

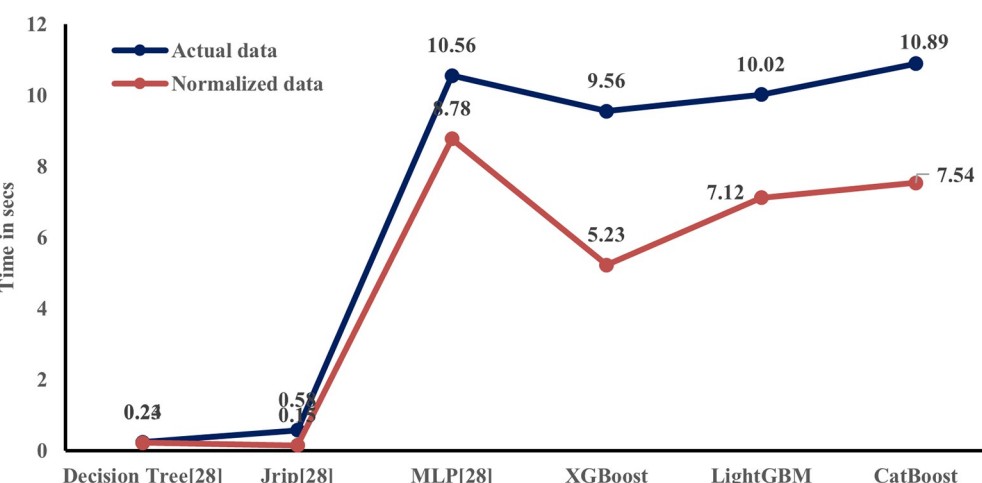

**Fig 10. Comparison of time requirements to build the model using an actual and normalized dataset.**

**Table 6. One-way ANOVA test of accuracy for CatBoost, XGBoost and LightGBM with rainfall and temperature feature sets.**

| Dataset | Algorithm | Mean±Standard Deviation | $p$-value |
|---|---|---|---|
| 'Temperature | CatBoost | 21.64±8.05 | 0.00027 |
| | LightGBM | 20.54±7.051 | |
| | CatBoost | 21.64±8.05 | 0.00038 |
| | XGBoost | 19.48±6.807 | |
| Rainfall | CatBoost | 1248.05±7.5 | 0.00049 |
| | LightGBM | 1149.05±7.05 | |
| | CatBoost | 1248.05±7.5 | 0.00053 |
| | XGBoost | 1050.060±6.90 | |

effectiveness with a crop yield dataset is assessed using its quantitative performance metrics. In accordance with the outcomes shown using the temperature and rainfall feature set, the Cat-Boost technique is superior to LightGBM and XGBoost.

As shown in Table 6, the test rejects the null hypothesis($H_0$) with $p$-values less than the 0.05 for each method. The ANOVA test indicated that there are significant differences between methods for both the rainfall and temperature feature sets($p < 0.05$).

We have shown that the CatBoost algorithm produces the best classification results using both combinations of feature sets (Rainfall and Temperature) in the Crop Yield dataset.

## Section 4: Conclusions and future scope

This paper delves into the efficacy of gradient methods in suggesting optimal crops for cultivation based on diverse attributes and environmental factors. The study reveals that all three gradient methods utilized yield precise outcomes, with CatBoost showcasing superior performance, boosting a coefficient of determination ($R^2$) of 0.964 and an impressive prediction accuracy of 99.1%. Despite the commendable accuracy, it is noted that these gradient methods demand a relatively longer duration for model construction.

### CatBoost dominance

The results underscore CatBoost as the most effective gradient method, achieving a remarkable $R^2$ of 0.964 and an accuracy rate of 99.1%. While the other gradient methods deliver comparable results, they exhibit prolonged model-building times.

### Future directions

**Integrated framework.** The study suggests the potential development of an integrated framework merging insights from IoT platforms and Artificial Intelligence (AI) to create a more sophisticated application for farmers. This integrated system could leverage available information and knowledge for enhanced decision-making.

**Expansion to soil fertility.** Future iterations of the application could incorporate additional factors such as soil fertility to provide a more comprehensive recommendation system for crop cultivation.

**Continuous environmental monitoring.** The envisioned application could be extended to continually monitor environmental conditions. This would enable the system to deliver timely alerts to farmers, safeguarding their crops against potential threats.

## Conclusion

In conclusion, this research demonstrates the robust capabilities of gradient methods, especially CatBoost, in recommending suitable crops based on diverse attributes. The remarkable accuracy achieved provides a solid foundation for the development of smarter applications that can significantly assist farmers in optimizing their cultivation decisions. The integration of IoT and AI, coupled with considerations for soil fertility, represents a promising avenue for future advancements in precision agriculture. Continuous environmental monitoring further enhances the application's utility, ensuring proactive protection of crops.

## Author Contributions

**Conceptualization:** Pavithra Mahesh.

**Data curation:** Pavithra Mahesh.

**Formal analysis:** Rajkumar Soundrapandiyan.

**Investigation:** Rajkumar Soundrapandiyan.

**Methodology:** Pavithra Mahesh.

**Project administration:** Rajkumar Soundrapandiyan.

**Resources:** Pavithra Mahesh.

**Supervision:** Rajkumar Soundrapandiyan.

**Writing – original draft:** Pavithra Mahesh.

**Writing – review & editing:** Pavithra Mahesh, Rajkumar Soundrapandiyan.

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
