## [Decision Letter · Decision Letter 0]

4 Jul 2023

PONE-D-23-17460Yield Prediction for Crops by Gradient Based AlgorithmsPLOS ONE

Dear Dr. Soundrapandiyan,

Thank you for submitting your manuscript to PLOS ONE. After careful consideration, we feel that it has merit but does not fully meet PLOS ONE’s publication criteria as it currently stands. Therefore, we invite you to submit a revised version of the manuscript that addresses the points raised during the review process.

ACADEMIC EDITOR: Dear Authors,

please revise proposed manuscript thoroughly according to all reviewers' comments.Additionally, my own comments are provided below.All the best,AE

We look forward to receiving your revised manuscript.

Kind regards,

Nebojsa Bacanin

Academic Editor

PLOS ONE

Journal Requirements:

Additional Editor Comments:

Dear Authors,

please revise proposed manuscript thoroughly according to all reviewers' comments.

Additionally, please do the following:

- Visualization of obtained results must be improved.

- Motivation behind proposed research should be more clearly explain. Please elaborate what is "beyond state-of-the-art" of proposed. study.

- To prove the significance of obtained results, statistical tests must be conducted. There are many statistical tests appropriate for validating results, please choose some tests from the following reference: https://www.sciencedirect.com/science/article/pii/S2210650211000034

- For the sake of clarity, best obtained metrics in each table should be marked e.g. by using the bold style.

- Make sure that the source code is available according to PLOS ONE publication policies.

Reviewers' comments:

Reviewer's Responses to Questions

**Comments to the Author**

1. Is the manuscript technically sound, and do the data support the conclusions?

Reviewer #1: Yes

Reviewer #2: Partly

2. Has the statistical analysis been performed appropriately and rigorously? 

Reviewer #1: Yes

Reviewer #2: No

3. Have the authors made all data underlying the findings in their manuscript fully available?

Reviewer #1: Yes

Reviewer #2: Yes

4. Is the manuscript presented in an intelligible fashion and written in standard English?

Reviewer #1: No

Reviewer #2: Yes

5. Review Comments to the Author

Reviewer #1: This manuscript evaluates various machine learning algorithms used to predict product performance. The authors compare the results of three machine learning algorithms, CatBoost, LightGBM and XGBoost. Their results show that CatBoost has the highest accuracy in yield prediction.

I went through the manuscript carefully. At the detailed level, the following notes are my suggestions:

1) Although the ABSTRACT structure is good, I suggest removing the first 5-6 lines and the last 3-4 lines. They are not really the descriptions that the reader expects to see in the ABSTRACT.

2) In my opinion, the INTRODUCTION section needs to be revised. In this section there should be three points: 1) motivation, 2) a summary of the challenges of previous studies, and 3) contribution. Also, the research contributions should be mentioned in a bullet-form at the end of the INTRODUCTION.

3) The INTRODUCTION section is too long. It is almost 10 pages! Instead, authors should reduce the INTRODUCTION to 2 pages. Then add a section called RELATED WORKS that provides a summary of previous studies. Also having subsection 1-1 doesn't help. Merge it with Section 1.

4) It is not clear to me which formulas were invented by the authors themselves and which ones are derived from other references. I found evidence that some formulas are derived from other references and there are similarities.

5) Authors should also use common supervised learning metrics such as accuracy, precision, and recall. For this purpose, I recommend adding the following reference and using the definitions of the above metrics from there:

https://www.tandfonline.com/doi/abs/10.1080/0952813X.2022.2153279

6) There are still some grammatical errors in the manuscript. Authors should use software such as Grammarly for proof-checking.

7) The tense of the verbs in the CONCLUSION section must be past tense. In this section, the most important numerical improvements of the proposed method should be mentioned and marginal explanations should be avoided. Also, the suggestions mentioned for further research should be presented in a new paragraph.

Reviewer #2: 1. Try to avoid using acronyms in the abstract.

2. Introduction is too long, please reduce it to max two pages.

3. Provide additional section - Background and related work, where you should provide literature survey.

4. Literature background must be expanded, as it is very limited. Include more recent relevant papers dealing with metaheuristics-based models, to provide stronger background on ML models. Include the following:

https://www.mdpi.com/2305-6304/11/4/394

https://www.sciencedirect.com/science/article/abs/pii/S2352710222010555

https://peerj.com/articles/cs-956/

https://ieeexplore.ieee.org/abstract/document/9840700

https://link.springer.com/chapter/10.1007/978-981-19-7753-4_60

https://www.atlantis-press.com/proceedings/iciitb-22/125984192

5. Figures should be provided in much better quality.

6. Elaborate in more details why gradient boosting methods were selected for this task, among other ML models.

7. The proposed model should be explained in more details.

8. There are some errors in equations (square signs where operators should be).

9. Make sure that each parameter in every equation has been explained in the text.

10. Elaborate in more details how the hyperparameters for the models were selected.

11. Finally, I would recommend performing a statistical analysis, to establish if the obtained results are statistically significant in comparison to other methods.

12. Paper needs extensive proofreading, as there are numerous grammatical and language errors in the text.

6. PLOS authors have the option to publish the peer review history of their article (what does this mean?). If published, this will include your full peer review and any attached files.

Reviewer #1: No

Reviewer #2: No

---

## [Author Response · Author response to Decision Letter 0]

6 Sep 2023

• A rebuttal letter that responds to each point raised by the academic editor and reviewer(s). You should upload this letter as a separate file labeled 'Response to Reviewers.

Author Response: We would like to thank the reviewer for pointing out this. We have revised the paper presented in PLOS ONE Format

2. Please note that PLOS ONE has specific guidelines on code sharing for submissions in which author-generated code underpins the findings in the manuscript. In these cases, all author-generated code must be made available without restrictions upon publication of the work. Please review our guidelines at https://journals.plos.org/plosone/s/materials-and-software-sharing#loc-sharing-code and ensure that your code is shared in a way that follows best practices and facilitates reproducibility and reuse.

Autor Response: Noted and the code will be shared after receiving acceptance for the article for publication 

b) State what role the funders took in the study. If the funders had no role in your study, please state: “The funders had no role in study design, data collection, and analysis, decision to publish, or preparation of the manuscript.”

Autor Response: No funding has been received for the study. We have mentioned this in the paper as advised.

Author Response: The dataset used for this research is publicly and freely available on the internet. We described data availability in the cover letter.

Autor Response: The corresponding author (Rajkumar S ) has an ORCID iD: 0000-0001-5701-9325

Additionally, please do the following:

-Visualization of obtained results must be improved.

Autor Response: We have improved visualization of obtained results in the revised manuscript.

- Motivation behind proposed research should be more clearly explain. 

Author Response: We would like to thank the reviewer for raising this important point in the motivation of the study. We have enhanced the motivation of the study where it reads; [#page 4]

- Please elaborate what is "beyond state-of-the-art" of proposed. study.

Author Response: The main intent of the study is to verify the effectiveness of the gradient methods and the algorithms are finely modified to avoid overfitting and underfitting. The same is mentioned in the Introduction.

- To prove the significance of obtained results, statistical tests must be conducted.

- There are many statistical tests appropriate for validating results, please choose some tests from the following reference: https://www.sciencedirect.com/science/article/pii/S2210650211000034

- For the sake of clarity, best obtained metrics in each table should be marked e.g. by using the bold style.

- Make sure that the source code is available according to PLOS ONE publication policies.

-

Author Response: We would like to thank the reviewer for raising this important point .As per reviewer comments we have conducted one way ANOVA statistical test to prove the significants of obtained results. [Page#21-22].

Comments to the Author

1. Is the manuscript technically sound, and do the data support the conclusions?

Reviewer #1: Yes

Reviewer #2: Partly

2. Has the statistical analysis been performed appropriately and rigorously?

Reviewer #1: Yes

Reviewer #2: No

3. Have the authors made all data underlying the findings in their manuscript fully available?

Reviewer #1: Yes

Reviewer #2: Yes

4. Is the manuscript presented in an intelligible fashion and written in standard English?

Reviewer #1: No

Reviewer #2: Yes

5. Review Comments to the Author

Reviewer #1: This manuscript evaluates various machine learning algorithms used to predict product performance. The authors compare the results of three machine learning algorithms, CatBoost, LightGBM and XGBoost. Their results show that CatBoost has the highest accuracy in yield prediction.

I went through the manuscript carefully. At the detailed level, the following notes are my suggestions:

1) Although the ABSTRACT structure is good, I suggest removing the first 5-6 lines and the last 3-4 lines. They are not really the descriptions that the reader expects to see in the ABSTRACT.

Author Response: We would like to thank the reviewer for raising this important point in the abstract of the study. We have enhanced the abstract of the study where it reads;

Abstract: A timely and consistent assessment of crop yield will assist the farmers in improving their income, minimizing losses, and deriving strategic plans in agricultural commodities to adopt import-export policies. Crop yield predictions are one of the various challenges faced in the agriculture sector and play a significant role in planning and decision-making. Machine learning algorithms provided enough belief and proved their ability to predict crop yield. The selection of the most suitable crop is influenced by various environmental factors such as temperature, soil fertility, water availability, quality, and seasonal variations, as well as economic considerations such as stock availability, preservation capabilities, market demand, purchasing power, and crop prices. The paper outlines a framework used to evaluate the performance of various machine-learning algorithms for forecasting crop yields. The models were based on a range of prime parameters including pesticides, rainfall & avg. temperature. The Results of three machine learning algorithms, Categorical Boosting (CatBoost), Light Gradient-Boosting Machine LightGBM, and Extreme Gradient Boosting XGBoost are compared and found more accurate than other algorithms in predicting crop yields. The RMSE and R2 values were calculated to compare the predicted and observed rice yields, resulting in the following values: Cat-Boost with 800 (0.24), LightGBM with 737 (0.33), and XGBoost with 744 (0.31). Among these three machine learning algorithms, CatBoost demonstrated the highest precision in predicting yields, achieving an accuracy rate of 99.123%. 

2) In my opinion, the INTRODUCTION section needs to be revised. In this section there should be three points: 1) motivation, 2) a summary of the challenges of previous studies, and 3) contribution. Also, the research contributions should be mentioned in a bullet-form at the end of the INTRODUCTION.

Author Response: We would like to thank the reviewer for raising this important point in the motivation of the study. We have enhanced the motivation of the study where it reads; [Page#4]

Motivation of the study is to develops an effective machine learning-based approach to crop yield prediction. Model a Prediction tool based on accurate crop yield predictions that can assist farmers and decision-makers about crop management, resource allocation, and risk mitigation strategies. With the uncertain weather patterns due to global warming, such a tool can be particularly useful in helping farmers adapt to changing conditions and ensure food security for the growing population.

Author Response: Summary of the challenges of previous studies has been included in the revised manuscript [Page#3]

Author Response: Contribution of the research have been mentioned in bullet-form at the end of the Introduction [Page#5]

Author Response: The main contribution of the work presented in this paper is outlined below

• Leveraging publicly available data on weather, agricultural practices, pesticides, and chemicals, a predictive model capable of accurately forecasting crop yields in India have been developed .

• One-Hot Encoding has been used to convert categorical variables to the one-hot numeric array

• Three different machine learning algorithms (CatBoost, LightGBM, and XGBoost) has been adopted in the model for achieving accurate prediction results for crops

• The developed robust prediction framework has been modelled to effectively avoid overfitting and underfitting scenarios.

3) The INTRODUCTION section is too long. It is almost 10 pages! Instead, authors should reduce the INTRODUCTION to 2 pages. Then add a section called RELATED WORKS that provides a summary of previous studies. Also having subsection 1-1 doesn't help. Merge it with Section 1.

Author Response: We would like to thank the reviewer for raising this important point in the introduction of the study. We have to reduce the introduction section in two pages [Page#2]

Author Response: We have included the following articles in the related works [Page#5].

[9]Jeong JH, Resop JP, Mueller ND, Fleisher DH, Yun K, et al. (2016) Random Forests for Global and Regional Crop Yield Predictions. PLOS ONE 11(6): e0156571. https://doi.org/10.1371/journal.pone.0156571

[13]Jovanovic, G.; Perisic, M.; Bacanin, N.; Zivkovic, M.; Stanisic, S.; Strumberger, I.; Alimpic, F.; Stojic, A. Potential of Coupling Metaheuristics-Optimized-XGBoost and SHAP in Revealing PAHs Environmental Fate. Toxics 2023, 11, 394. https://doi.org/10.3390/toxics11040394

[14] Jui-Sheng Chou, Chi-Yun Liu, Handy Prayogo, Riqi Radian Khasani, Danny Gho, Gretel Gaby Lalitan, Predicting nominal shear capacity of reinforced concrete wall in building by metaheuristics-optimized machine learning, Journal of Building Engineering, Volume 61, 2022, https://doi.org/10.1016/j.jobe.2022.105046. 

[15] Zivkovic M, Tair M, K V, Bacanin N, Hubálovský Š, Trojovský P. 2022. Novel hybrid firefly algorithm: an application to enhance XGBoost tuning for intrusion detection classification. PeerJ Computer Science 8:e956 https://doi.org/10.7717/peerj-cs.956

[16] A. Petrovic, I. Strumberger, M. Antonijevic, D. Jovanovic, D. Mladenovic and A. Chabbra, "Firefly-Xgboost Approach for Pedestrian Detection," 2022 IEEE Zooming Yield Prediction for Crops by Gradient Based Algorithms 19 / 19 Innovation in Consumer Technologies Conference (ZINC), Novi Sad, Serbia, 2022, pp. 197- 202, doi: 10.1109/ZINC55034.2022.9840700.

[17] Jovanovic, L. et al. (2023). Tuning XGBoost by Planet Optimization Algorithm: An Application for Diabetes Classification. In: Bindhu, V., Tavares, J.M.R.S., Vuppalapati, C. (eds) Proceedings of Fourth International Conference on Communication, Computing and Electronics Systems. Lecture Notes in Electrical Engineering, vol 977. Springer, Singapore. https://doi.org/10.1007/978-981-19-7753-4_60

[18] Petrovic, A., Antonijevic, M., Strumberger, I., Jovanovic, L., Savanovic, N., & Janicijevic, S. (2023, January). The XGBoost Approach Tuned by TLB Metaheuristics for Fraud Detection. In Proceedings of the 1st International Conference on Innovation in Information Technology and Business (ICIITB 2022) (Vol. 104, p. 219). Springer Nature.

[23] Shook J, Gangopadhyay T, Wu L, Ganapathysubramanian B, Sarkar S, et al. (2021) Crop yield prediction integrating genotype and weather variables using deep learning. PLOS ONE 16(6): e0252402. https://doi.org/10.1371/journal.pone.0252402

4) It is not clear to me which formulas were invented by the authors themselves and which ones are derived from other references. I found evidence that some formulas are derived from other references and there are similarities.

Author Response: The formulae are adopted from reference papers.[not invented by authors] The basic equations are similar/same and will not change. However we have framed objective function from the basic equations to suit crop prediction.

5) Authors should also use common supervised learning metrics such as accuracy, precision, and recall. For this purpose, I recommend adding the following reference and using the definitions of the above metrics from there:

https://www.tandfonline.com/doi/abs/10.1080/0952813X.2022.2153279

Author Response: We agree with the reviewer, the metric mentioned are commonly used. But We have adopted the metrics used in reference paper to compare the results. Hence kindly excuse for not adding the reference recommended.

6) There are still some grammatical errors in the manuscript. Authors should use software such as Grammarly for proof-checking.

Author Response: Thank you for pointing grammatical errors in our manuscript. We have carefully studied the manuscript and minimized the grammatical mistakes using Grammarly.

7) The tense of the verbs in the CONCLUSION section must be past tense. In this section,

---

## [Decision Letter · Decision Letter 1]

26 Oct 2023

PONE-D-23-17460R1Yield Prediction for Crops by Gradient-Based AlgorithmsPLOS ONE

Dear Dr. Soundrapandiyan,

Thank you for submitting your manuscript to PLOS ONE. After careful consideration, we feel that it has merit but does not fully meet PLOS ONE’s publication criteria as it currently stands. Therefore, we invite you to submit a revised version of the manuscript that addresses the points raised during the review process.

We look forward to receiving your revised manuscript.

Kind regards,

Nebojsa Bacanin

Academic Editor

PLOS ONE

**Additional Editor Comments:**

Dear Authors,

please revise your manuscript carefully according to reviewers' comments.

Thank you.

Reviewers' comments:

Reviewer's Responses to Questions

**Comments to the Author**

1. If the authors have adequately addressed your comments raised in a previous round of review and you feel that this manuscript is now acceptable for publication, you may indicate that here to bypass the “Comments to the Author” section, enter your conflict of interest statement in the “Confidential to Editor” section, and submit your "Accept" recommendation.

Reviewer #1: All comments have been addressed

Reviewer #2: All comments have been addressed

Reviewer #3: (No Response)

Reviewer #4: (No Response)

Reviewer #5: All comments have been addressed

2. Is the manuscript technically sound, and do the data support the conclusions?

Reviewer #1: Yes

Reviewer #2: (No Response)

Reviewer #3: No

Reviewer #4: Yes

Reviewer #5: Yes

3. Has the statistical analysis been performed appropriately and rigorously? 

Reviewer #1: Yes

Reviewer #2: (No Response)

Reviewer #3: No

Reviewer #4: Yes

Reviewer #5: Yes

4. Have the authors made all data underlying the findings in their manuscript fully available?

Reviewer #1: No

Reviewer #2: (No Response)

Reviewer #3: No

Reviewer #4: Yes

Reviewer #5: No

5. Is the manuscript presented in an intelligible fashion and written in standard English?

Reviewer #1: Yes

Reviewer #2: (No Response)

Reviewer #3: No

Reviewer #4: Yes

Reviewer #5: Yes

6. Review Comments to the Author

Reviewer #1: The authors appreciated my comments. I have no other comment. From my husband, this manuscript can be published.

Reviewer #2: (No Response)

Reviewer #3: The article presents a study on Yield Prediction for Crops by Gradient Based Algorithms. In comparison of various other studies published on crop prediction in other journals, I found that this article lacks novelty in terms of originality and technicality. The methods used for crop analysis is very trivial and outdated. Moreover, the results do not provide any useful information for agriculture domain as well. I would suggest a rejection.

Reviewer #4: 1. Introduction may be improved, adding the highlights and the problem statements.

2. You could improve writing, link better the ideas flow in the Introduction.

3. Review references because some of them are unstandardized.

4. The conclusion needs improvements towards major claimed contribution.

5. Write some future directions in the conclusion section.

Reviewer #5: The comparative assessment of different models may be statistically tested. Authors may use Diebold Mariano test for this purpose. you may see following paper https://doi.org/10.1371/journal.

pone.0270553

MAPE (mean absolute prediction error) may also be reported.

7. PLOS authors have the option to publish the peer review history of their article (what does this mean?). If published, this will include your full peer review and any attached files.

Reviewer #1: No

Reviewer #2: No

Reviewer #3: No

Reviewer #4: No

Reviewer #5: No

---

## [Author Response · Author response to Decision Letter 1]

7 Dec 2023

Response to Reviewers

Manuscript Number: PONE-D-23-17460R1

Title: Yield Prediction for Crops by Gradient-Based Algorithms

PLOS ONE

Dear editors and reviewers,

Thank you for consideration and for your efforts to bring the work in better way. We hereby submit a revised version of the manuscript that addresses the points raised during the review process.

Please find below our response to each point raised by the academic editor and reviewer(s). 

We here by submit the below documents along with response to reviewers:

A separate file labelled 'Revised Manuscript with Track Changes' is uploaded. The marked-up copy of your manuscript that highlights changes made to the original version. 

A separate file labelled 'Manuscript' An unmarked version of the revised paper without tracked changes. 

1. If the authors have adequately addressed your comments raised in a previous round of review and you feel that this manuscript is now acceptable for publication, you may indicate that here to bypass the “Comments to the Author” section, enter your conflict-of-interest statement in the “Confidential to Editor” section, and submit your "Accept" recommendation.

Reviewer #1: All comments have been addressed

Reviewer #2: All comments have been addressed

Reviewer #3: (No Response)

Reviewer #4: (No Response)

Reviewer #5: All comments have been addressed

Author Response: No response required for the comment, since reviewers have concurred for addressing the comments 

2. Is the manuscript technically sound, and do the data support the conclusions?

 Reviewer #1: Yes

Reviewer #2: (No Response)

Reviewer #3: No

Reviewer #4: Yes

Reviewer #5: Yes

Author Response: Performance Evaluation of Catboost, XGBoost, and LightGBM by Diebold-Mariano test for comparison has been additionally included and except one reviewer, other feel the results presented in the paper is adequate.

 3. Has the statistical analysis been performed appropriately and rigorously?

 Reviewer #1: Yes

Reviewer #2: (No Response)

Reviewer #3: No

Reviewer #4: Yes

Reviewer #5: Yes

Author Response: Statistical analysis by Diebold-Mariano test has been additionally included and except one reviewer, other feel the results presented in the paper is adequate.

 4. Have the authors made all data underlying the findings in their manuscript fully available?

 Reviewer #1: No

Reviewer #2: (No Response)

Reviewer #3: No

Reviewer #4: Yes

Reviewer #5: No

Author Response: Dataset obtained from third party used for experimentation has been referred in the paper [24] and results obtained during experimentation have been recorded fully in the paper. If information in specific is required, kindly mention. The authors are willing to send.

 5. Is the manuscript presented in an intelligible fashion and written in standard English?

 Reviewer #1: Yes

Reviewer #2: (No Response)

Reviewer #3: No

Reviewer #4: Yes

Reviewer #5: Yes

Author Response: It can be noted the reviewers (except Reviewer #3) considered the paper is written in standard English and in better manner. Considering the Reviewer #3 response, we have made few corrections for better understanding and reading.

6. Review Comments to the Author

 Reviewer #1: The authors appreciated my comments. I have no other comment. From my husband, this manuscript can be published.

Reviewer #2: (No Response)

Reviewer #3: The article presents a study on Yield Prediction for Crops by Gradient Based Algorithms. In comparison of various other studies published on crop prediction in other journals, I found that this article lacks novelty in terms of originality and technicality. The methods used for crop analysis is very trivial and outdated. Moreover, the results do not provide any useful information for agriculture domain as well. I would suggest a rejection.

Author Response: Kindly note, our intent is to investigate the capability of gradient methods for crop predictions. The experimentation results provide the proof for capacity of gradient methods. The results give us confident to enhance the model considering below aspects to reach a complete frame work.

• Integrated Framework Development

• Incorporation of Additional Environmental Factors

• Enhanced Predictive Models

• Continuous Environmental Monitoring System

• Exploration of Multi-Crop Recommendations

• Integration with Precision Agriculture Technologies

• Community Engagement and Farmer Education

• Long-Term Impact Assessment

Reviewer #4: 1. Introduction may be improved, adding the highlights and the problem statements.

Author Response: We would like to thank the reviewer for pointing out this. Problem statement with highlights of the work included in introduction of the revised paper [Page#3].

2. You could improve writing, link better the ideas flow in the Introduction.

Author Response: We would like to thank the reviewer for raising this important point in the introduction, the paper starts with aspects of framing, problem statement (now included), continued with a brief about IOT and data Analysis, motivation of the study, contributions and content covered in sessions of the paper. A paragraph of details to link the flow of ideas has been included.

3. Review references because some of them are unstandardized.

Author Response: We would like to thank the reviewer for raising this important point. As per reviewer comments we have included references from standard journals and revised the paper [Page#28-31].

4. The conclusion needs improvements towards major claimed contribution.

Author Response: As per the reviewer’s suggestion we have improved the conclusion toward major claimed contribution. [Page#26-27]

5. Write some future directions in the conclusion section.

Author Response: As per the suggestion of the reviewer the future directions have been included in conclusion section. [Page#26-27]

Reviewer #5: The comparative assessment of different models may be statistically tested. Authors may use Diebold Mariano test for this purpose. you may see following paper https://doi.org/10.1371/journal.

pone.0270553

MAPE (mean absolute prediction error) may also be reported.

Author Response: Thank you for raising this important comment. The comparative assessment of different models has been statistically tested by Diebold-Mariano test in revised paper [Page# 21-22]. Moreover, as per reviewer’s suggestion, we have included mean absolute percentage error [Page#19-20].

 7. PLOS authors have the option to publish the peer review history of their article (what does this mean?). If published, this will include your full peer review and any attached files.

Do you want your identity to be public for this peer review? For information about this choice, including consent withdrawal, please see our Privacy Policy.

 Reviewer #1: No

Reviewer #2: No

Reviewer #3: No

Reviewer #4: No

Reviewer #5: No

Author Response: Noted

---

## [Decision Letter · Decision Letter 2]

1 Feb 2024

Yield Prediction for Crops by Gradient-Based Algorithms

PONE-D-23-17460R2

Dear Dr. Soundrapandiyan,

We’re pleased to inform you that your manuscript has been judged scientifically suitable for publication and will be formally accepted for publication once it meets all outstanding technical requirements.

Kind regards,

Nebojsa Bacanin

Academic Editor

PLOS ONE

Additional Editor Comments (optional):

Dear Authors,

thank you for revising your manuscript.

Warmest,

AE

Reviewers' comments:

Reviewer's Responses to Questions

**Comments to the Author**

1. If the authors have adequately addressed your comments raised in a previous round of review and you feel that this manuscript is now acceptable for publication, you may indicate that here to bypass the “Comments to the Author” section, enter your conflict of interest statement in the “Confidential to Editor” section, and submit your "Accept" recommendation.

Reviewer #4: All comments have been addressed

Reviewer #5: All comments have been addressed

2. Is the manuscript technically sound, and do the data support the conclusions?

Reviewer #4: Yes

Reviewer #5: Yes

3. Has the statistical analysis been performed appropriately and rigorously? 

Reviewer #4: Yes

Reviewer #5: Yes

4. Have the authors made all data underlying the findings in their manuscript fully available?

Reviewer #4: Yes

Reviewer #5: Yes

5. Is the manuscript presented in an intelligible fashion and written in standard English?

Reviewer #4: Yes

Reviewer #5: (No Response)

6. Review Comments to the Author

Reviewer #4: All my concerns have been addressed.

In the revised version of the manuscript, the authors met all the requirements and comments given in the previous review, so I recommend this paper for publishing.

Reviewer #5: The suggestions have been incorporated by the authors in the revised manuscript. The paper is improved signfiicantly

7. PLOS authors have the option to publish the peer review history of their article (what does this mean?). If published, this will include your full peer review and any attached files.

Reviewer #4: No

Reviewer #5: No

---

## [Editor Report · Acceptance letter]

29 Mar 2024

PONE-D-23-17460R2 

PLOS ONE

Dear Dr. Soundrapandiyan, 

I'm pleased to inform you that your manuscript has been deemed suitable for publication in PLOS ONE. Congratulations! Your manuscript is now being handed over to our production team.

Kind regards, 

on behalf of

Dr. Nebojsa Bacanin 

Academic Editor

PLOS ONE